nanotechnology/physical chemistry/computer modelling and simulation

ruptured, intermolecular hydrogen bond, glycerol, viscosity, graphite, graphene

**Authors for correspondence:**
Guoliang Zhang
e-mail: zgl19847985@163.com
Liran Ma
e-mail: maliran@mail.tsinghua.edu.cn

# Intermolecular hydrogen bond ruptured by graphite with different lamellar number

Yanchao Yin[1], Guoliang Zhang[2], Xianmang Xu[1], Peiyu Zhao[1] and Liran Ma[3]

[1]Heze Branch, Biological Engineering Technology Innovation Center of Shandong Province, Qilu University of Technology (Shandong Academy of Sciences), Heze 274000, Shandong, People's Republic of China
[2]School of Mechanical Engineering, Tianjin University of Technology and Education, Tianjin 300222, People's Republic of China
[3]State Key Laboratory of Tribology, Tsinghua University, Beijing 100084, People's Republic of China

YY, 0000-0002-5037-0435

Intermolecular hydrogen bonds are formed through the electrostatic attraction between the hydrogen nucleus on a strong polar bond and high electronegative atom with an unshared pair of electrons and a partial negative charge. It affects the physical and chemical properties of substances. Based on this, we presented a physical method to modulate intermolecular hydrogen bonds for not changing the physical–chemical properties of materials. The graphite and graphene are added into the glycerol, respectively, by being used as a viscosity reducer in this paper. The samples are characterized by Raman and 1H-nuclear magnetic resonance. Results show that intermolecular hydrogen bonds are adjusted by graphite or graphene. The rheology of glycerol is reduced to varying degrees. Transmission electron microscopes and computer simulation show that the spatial limiting action of graphite or graphene is the main cause of breaking the intermolecular hydrogen bond network structure. We hope this work reveals the potential interplay between nanomaterials and hydroxyl liquids, which will contribute to the field of solid–liquid coupling lubrication.

# 1. Introduction

Intermolecular hydrogen bond was presented by the IUPAC in 2004. It was defined [1] that the hydrogen atom is covalently bonded to the atom X with large electronegativity. If it is too close to the atom Y (O, F, N, etc.) with large electronegativity and small radius, hydrogen is used as the medium between X and Y to generate X–H…Y type of intermolecular or

This article has been edited by the Royal Society of Chemistry, including the commissioning, peer review process and editorial aspects up to the point of acceptance.

intramolecular interaction. Many researchers are attracted to study it. Li *et al*. [2] found that there were two stage temperature ranges in the process of polyurea intermolecular hydrogen bond failure. First, the bidentate hydrogen bond was dissociation in the temperature range of 85–165°C. When the temperature went over 165°C, intermolecular hydrogen bond was completely broken. Wang *et al*. [3] studied the impact of intermolecular hydrogen bond on the melting point of azole explosives. It found that hydrogen bonds influenced crystal packing modes, which also changed the entropy and enthalpy in melting process. Lim *et al*. [4] introduced that the increase in intermolecular hydrogen bond strength decreased the crystallization rates of the PHB–HH$x$/SiO$_2$ hybrids. So, the physical and chemical properties of materials are affected by intermolecular hydrogen bonding [5–8].

Based on the above review of intermolecular hydrogen bonds, this paper tries to introduce some nanomaterials to investigate destroying the intermolecular hydrogen bond network structure and its effect on the physical–chemical properties of substances. Therefore, graphene and graphite, being a single graphite layer structure and a multi-graphite layer structure, respectively, are used. They are used in refractory, metallurgical casting, electric conduction, lubrication and other fields [9–12]. Bag *et al*. [13] studied the effect of graphite content on the refractory of MgO–C. When the content of graphite was 3%, the refractory of MgO–C was better than that of traditional MgO–C. Mukhopadhyay *et al*. [14] found that the physicochemistry property of the refractory containing graphite was superior to that of the refractory without graphite. Akhlaghi *et al*. [15] provided aluminium alloy–graphite composites by *in situ* powder metallurgy. The addition of graphite obviously improved the wear property of Al-alloy. Dou *et al*. [16] found that the Zn-doped SnO$_2$ nanospheres coated with graphene had a larger specific surface, greater electrical conductivity and a longer cycle life. Chen *et al*. [17] reported that single-layer graphene had unique conductive effect under high electric field, which was called the current self-amplification effect. Mishra *et al*. [18] studied the wear damage at very high contact pressure. The results showed that graphene-based lubricant could suppress the subsurface cracking and delamination in high contact pressure, while the oil-based lubricant could not. Ravindran *et al*. [19] investigated the tribological behaviour of Al-2024–graphite composites. The results showed that the friction and wear properties of the composites with 5 wt% graphite were obviously improved. This was due to the fact that graphite is also a good solid lubricant.

Excellent physicochemical properties of graphene and graphite gave them good solid lubricating property. In this paper, we try to add them to glycerol, respectively, and use their spatial limiting action after dispersion to destroy the hydrogen bond between glycerol molecules, while retaining the lubrication property of their solid lubricant. The ability of adjusting the intermolecular hydrogen bonds of glycerol after graphite and graphene addition was analysed, and the modulation mechanism was established. Meanwhile, the rheological properties of the mixture, after intermolecular hydrogen bonding was adjusted, were investigated.

# 2. Experimental procedure

Graphite nanoparticles (Nanjing XFNANO Materials Tech Co., Ltd) were mixed with glycerol through ultrasound for a certain time at 25 ± 1°C. Then, the mixture was set for greater than 16 h. Glycerol mixed with graphene (The Six Element Inc.) was also prepared in the same way. The mixing mass ratio, respectively: $m$(glycerol)/$m$(graphite) and $m$(glycerol)/$m$(graphene) were denoted, respectively, by $x$ and $y$ ($x$, $y$ = 500, 750, 1000, 1250, 1500).

The mixed state of sample was analysed by radio mode of Horiba LabRAM HR, and the resolution was 2 µm. $^1$H-nuclear magnetic resonance (NMR) was carried out using dimethyl sulfoxide-d6 as solvent and configuring 0.5 mol l$^{-1}$ detection solution concentration. The rheology of the mixture was evaluated on Anton Paar Physica MCR301 under air-bearing torque of 0.002 µN m to 200 mN m and the detection temperature was 25 ± 1°C. The microscopic distribution of graphite and graphene was determined by JEM-2100 transmission electron microscopes (TEM). The computer simulation was performed at the same time.

# 3. Results and discussion

The interaction between substances was reflected by a shift in the Raman spectrum peak. The strength of the chemical bonds was judged by the red and blue shift of the Raman spectrum peak. The Raman spectrum peak of infiltrated graphite and graphene is shown in figure 1.

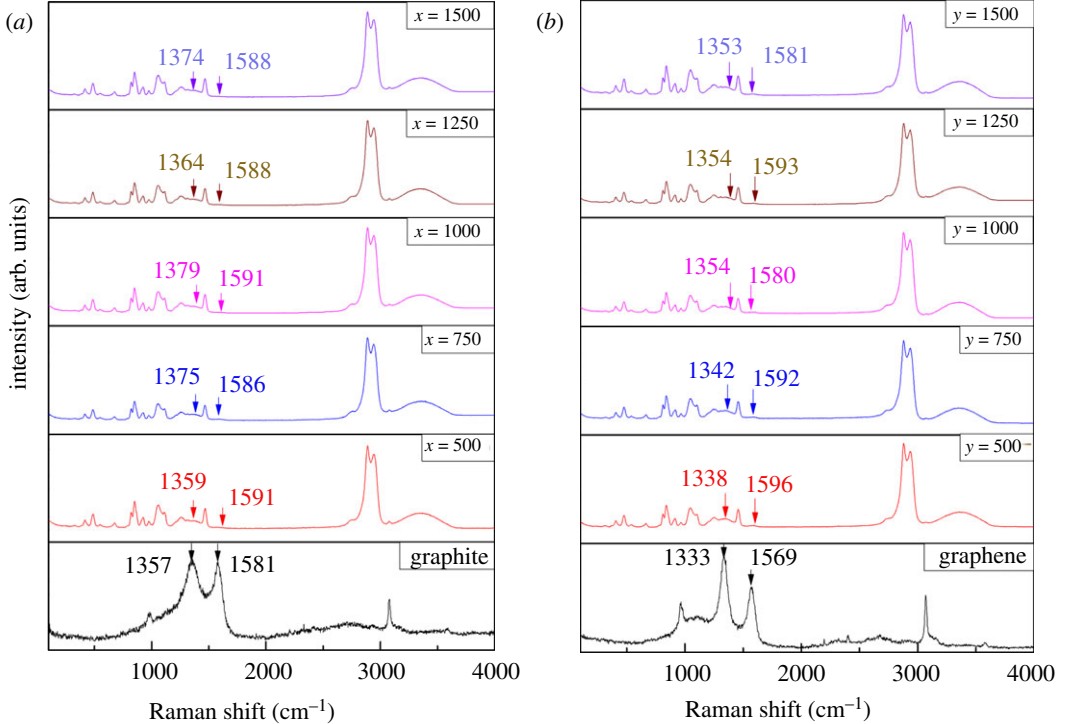

**Figure 1.** Raman spectrum of graphite and graphene in glycerol. (*a*) Raman spectrum peak of graphite and infiltrated graphite. After being infiltrated, a red shift occurred in the Raman spectrum peak of them. (*b*) Raman spectrum peak of graphene and infiltrated graphene. The red shift of Raman peak also occurred. Similarly, graphene had an interaction with glycerol molecules.

The Raman spectrum peaks of the carbon materials were near 1350 and 1580 cm$^{-1}$. As shown in figure 1, the peaks of pure graphite and pure graphene were 1357/1581 and 1333/1569 cm$^{-1}$, respectively. After being infiltrated, the Raman spectrum peak of graphite and graphene had different degrees of displacement (red shift). This illustrated that graphite and graphene interacted with glycerol molecules, respectively, and the bonding ability of hydrogen bonds between glycerol molecules became weaker.

From the size of the Raman spectrum peak displacement, the defect position Raman peak (approx. 1350 cm$^{-1}$) of the graphite clearly shifted, while the Raman peak (approx. 1580 cm$^{-1}$) of the graphene clearly shifted. Multi-layer structure and spheroidal structure of graphite resulted in more defect positions existing. Glycerol molecules were easier to interact with defect positions. However, single-layer structure of graphene had less defect positions. The Raman spectrum peak (approx. 1580 cm$^{-1}$) of graphene had a more obvious red shift because of the charge attraction and the strong hydrophobicity of graphene. The intermolecular hydrogen bonds were modified by the interaction between graphite or graphene and glycerol molecules. The spatial confinement effect played a main role due to their strong hydrophobicity.

In order to further investigate the interaction between glycerol molecules and graphite or graphene, $^{1}$H-NMR analysis of the samples was carried out. Learning from the glycerol $^{1}$H-NMR standard spectrum, its characteristic peak of the intermolecular hydroxyl hydrogen was near 4.4 ppm.

The peak of hydroxyl hydrogen was shifted after adding graphite, as shown in figure 2*a*. Compared with that of pure glycerol, the chemical shift of hydroxyl hydrogen moved in a large direction. It was suggested that the spin motion of hydroxyl hydrogen was weakened after adding graphite. An interaction must happen between glycerol molecules and graphite, which limited the spin of hydroxyl hydrogen.

As shown in figure 2*b*, the peak was also shifted after adding graphene. The chemical shift of hydroxyl hydrogen was larger than that of adding graphite. It was shown that the interaction between monolayer graphene and glycerol molecules was more stable than that of graphite. The spin motion of hydroxyl hydrogen was also weakened by graphene.

The rheology curve of the mixtures (25°C) is shown in figure 3*a*,*b*. As shown in figure 3*a*, the viscosity of the mixture was lower than that of pure glycerol. Initially, when the amount of graphite was the largest (*x* = 500), the graphite was in a state of uniform dispersion in glycerol, and the ability to modulate the intermolecular hydrogen bond of glycerol was the strongest. With the decrease of adding amount, the modulation effect was weakened. Because of the large specific surface energy and the occurrence of small-

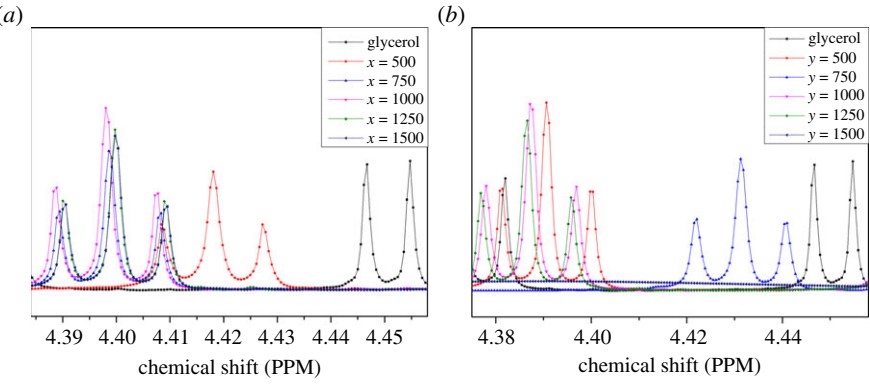

**Figure 2.** $^1$H-NMR of samples. (a) $^1$H-NMR spectrum peak of graphite and glycerol mixture. The peak of hydroxyl hydrogen had a blue shift after adding graphite. (b) $^1$H-NMR spectrum peak of graphene and glycerol mixture. The peak of hydroxyl hydrogen also had a blue shift after adding graphene.

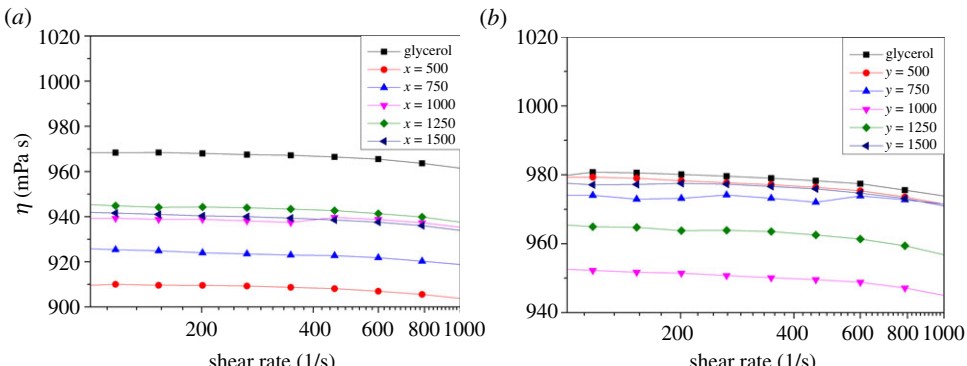

**Figure 3.** Rheological curves of samples. (a) Rheological behaviour of graphite and glycerol mixtures. The viscosity of glycerol was decreased by graphite. (b) Rheological behaviour of graphene and glycerol mixtures. The viscosity of glycerol was decreased by graphene.

scale agglomeration, when the amount of glycerol is reduced to $x = 1000/1250/1500$, its ability to modulate the intermolecular hydrogen bond of glycerol was basically the same. It showed that the thinning action of graphite was more dependent on its mass. Its action had been greatly weakened at $x = 1000$. Continuing to reduce its mass, the magnitude of viscosity reduction would not change significantly.

As shown in figure 3b, the thinning action of graphene was also obvious. With the decrease of the amount of graphene, the change law of viscosity of glycerol was different from that of graphite. It was easier for the surface of polar graphene to adsorb hydrocarbons. The more hydrocarbons adsorbed, the lower the surface energy, which reduced the agglomeration phenomenon. Compared with graphite, the number of monolayer graphene was much higher than that of graphite at the same amount. When the amount of graphene was the largest ($y = 500$), its aggregation in glycerol was serious, and its adsorption capacity for glycerol was weak. With the decrease of the amount of graphene, when the amount of graphene was reduced to $y = 1000$, the distribution of graphene in glycerol was more dispersed, and the adsorption of glycerol on each graphene reached saturation state. At this time, the graphene adsorbed by glycerol had lower specific surface energy. The aggregation phenomenon was reduced, and the ability to modulate the intermolecular hydrogen bond of glycerol was the strongest. However, with the decrease of the addition amount ($y = 1250/1500$), the modulation ability was weakened. This introduced that the thinning action of graphene had an optimal value, which could greatly reduce the viscosity of glycerol.

Usually, the physicochemical properties of graphite are not equal to that of graphene. The effect of multi-layer graphite on the viscosity of glycerol was more obvious than the effect of graphene in figure 3. In order to investigate this reason, infiltrated graphite and graphene were analysed by TEM. As shown in figure 4a,b, graphite and graphene were evenly distributed in glycerol. The distribution and size of graphite in glycerol were different before and after shearing. Multi-lamellar structure of graphite was destroyed by the higher shear rate (4000–7000 r.p.m.). After shearing, the dispersion density of graphite in glycerol was clearly increased. Graphite was also destroyed in the process of ultrasonic mixing. It can be seen clearly that some smaller graphite was found in glycerol in figure 4a.

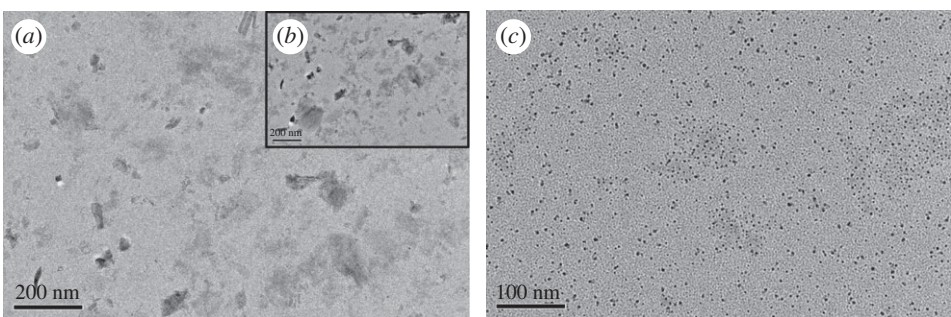

**Figure 4.** TEM images of infiltrated graphite and graphene. (*a*) The distribution of graphite in glycerol before shearing. Low dispersion density. (*b*) The distribution of graphite in glycerol after shearing. The dispersion density of graphite was improved clearly. (*c*) The distribution of graphene in glycerol. The distribution of graphene in glycerol was very uniform.

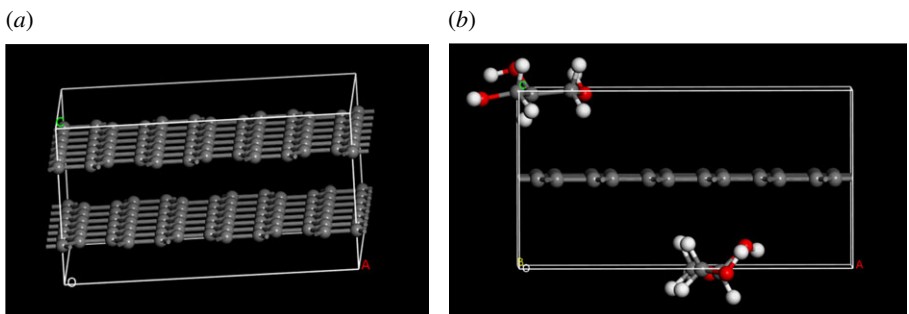

**Figure 5.** Adsorption model of graphite and graphene. The construction parameters of periodic single-layer graphene were that its supercell range was (6, 6, 1). Its lengths (Å) were $a = 12.6$, $b = 14.8$ and $c = 6.8$. Its angles were $\alpha = \beta = \gamma = 90°$. Black, red and white balls correspond to carbon, oxygen and hydrogen atoms, respectively. (*a*) Side view of graphite. It was not surrounded by glycerol molecules. (*b*) Side view of graphene. It was surrounded by some glycerol molecules.

Its space-limiting action became stronger. This damage increased the number of defect positions in the graphite, which explained why the Raman peak of graphite defect had shifted significantly.

Monolayer graphene was not destroyed by the higher shear rate in figure 4*c*. Its spatial confinement effect was not improved. Its thinning action on viscosity was mainly depending on its amount of addition. It was explained that the thinning ability of graphite on the viscosity of glycerol was stronger than that of graphene.

For more detailed description of the interaction between graphite and graphene with glycerol, a model of the adsorption of glycerol by graphite or graphene was established.

The model of graphene was introduced from MS database. The serrated supercell structure of graphene was (6*6*1).

The theory of partial seal transparency proposed by Shih *et al*. [20] and Li *et al*. [21] shows that the surface of pure monolayer graphene was polar with higher specific surface energy. The hydrocarbon was easy to be absorbed on its surface. The specific surface energy decreases with the increase of hydrocarbon adsorption. After adsorption and saturation, its hydrophobicity increases. As shown in figure 5*a*, there were no glycerol molecules around graphite according to the result of simulation. There were two glycerol molecules around graphene in figure 5*b*. This illustrated that graphite's multi-layer structure made its surface energy lower than that of monolayer graphene, and its hydrophobicity made it difficult to re-adsorb glycerol molecules on its surface. However, the space-limiting action of graphite and graphene in glycerol was still the main reason for the decrease of glycerol viscosity. Compared with monolayer graphene, graphite in glycerol could be destroyed at a high shear rate. Its multi-layer structure could be cut into even fewer layer or even single-layer structures. After shearing, the dispersion density of graphite was increased, which improved its space-limiting action. In addition, a small part of the newly formed monolayer graphene adsorbed glycerol molecules.

After graphite/graphene with glycerol was mixed completely, their space-limiting action provided a special space for glycerol molecules. These special spaces transformed large glycerol clusters into small clusters, reducing the number of intermolecular hydrogen bonds. The newly formed small glycerol clusters were isolated by graphite, which prevented the formation of intermolecular hydrogen bonds between clusters. Moreover, a small amount of glycerol molecules were adsorbed on the monolayer

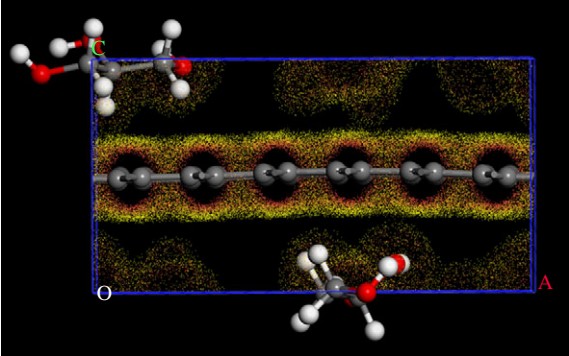

**Figure 6.** Differential charge density simulation of adsorption system. It was based on PBE functional in GGA by Castep model in MS software.

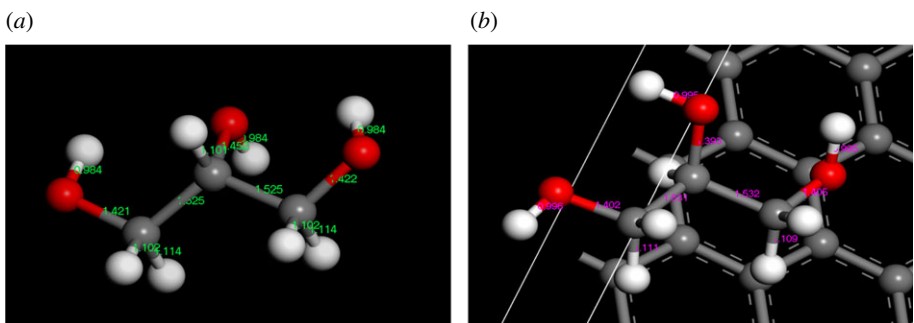

**Figure 7.** Changes of intramolecular bond length in glycerol. (*a*) Bond length before adsorption. (*b*) Bond length after adsorption.

**Table 1.** The length of C–O bond before and after adsorption.

| type of bond | bond length before adsorption (Å) | bond length after adsorption (Å) |
|---|---|---|
| C–O | 1.421 | 1.402 |
| C–O | 1.452 | 1.393 |
| C–O | 1.422 | 1.405 |

graphene surface, which also reduced the number of glycerol molecules and the number of intermolecular hydrogen bonds in the bulk phase. This caused the viscosity of glycerol to be reduced.

Modulation mechanism of intermolecular hydrogen bonds in this mixed system was simulated by the Castep model in MS software. The differential charge density simulation was based on Perdew–Burke–Ernzerhof (PBE) functional in generalized gradient approximation (GGA).

As shown in figure 6, the charge of glycerol molecule near monolayer graphene shifted to the surface of monolayer graphene, and then it was adsorbed and fixed on the surface of monolayer graphene. The attraction and migration of this charge led to the corresponding changes in the bond length of glycerol. The changes of bond length of glycerol before and after adsorption are shown in figure 7.

As shown in figure 7 and table 1, the C–O bond in glycerol molecule was shortened because of the shift of electrons. These changes made the spin motion of hydroxyl hydrogen become weak, which was consistent with the result of the [1]H-NMR analysis.

Therefore, the viscosity reduction mechanism of graphite and graphene in glycerol is shown in figure 8.

As shown in figure 8*a*, graphite was not surrounded by glycerol molecules due to its strong hydrophobicity. Multi-lamellar structure of graphite was destroyed in the process of ultrasonic and shear, which was in accordance with the results in figure 4*b*. The number of its layer was reduced, and even graphene was formed. The interaction between graphene and glycerol molecules is shown in figure 8*b*. Molecular clusters were broken by ultrasonic. At the same time, graphene was evenly dispersed in glycerol. A few glycerol molecules were absorbed on the surface of monolayer graphene and separated from each other by graphene. As shown in figure 8*a*, according to the simulation, only the monolayer

(*a*)

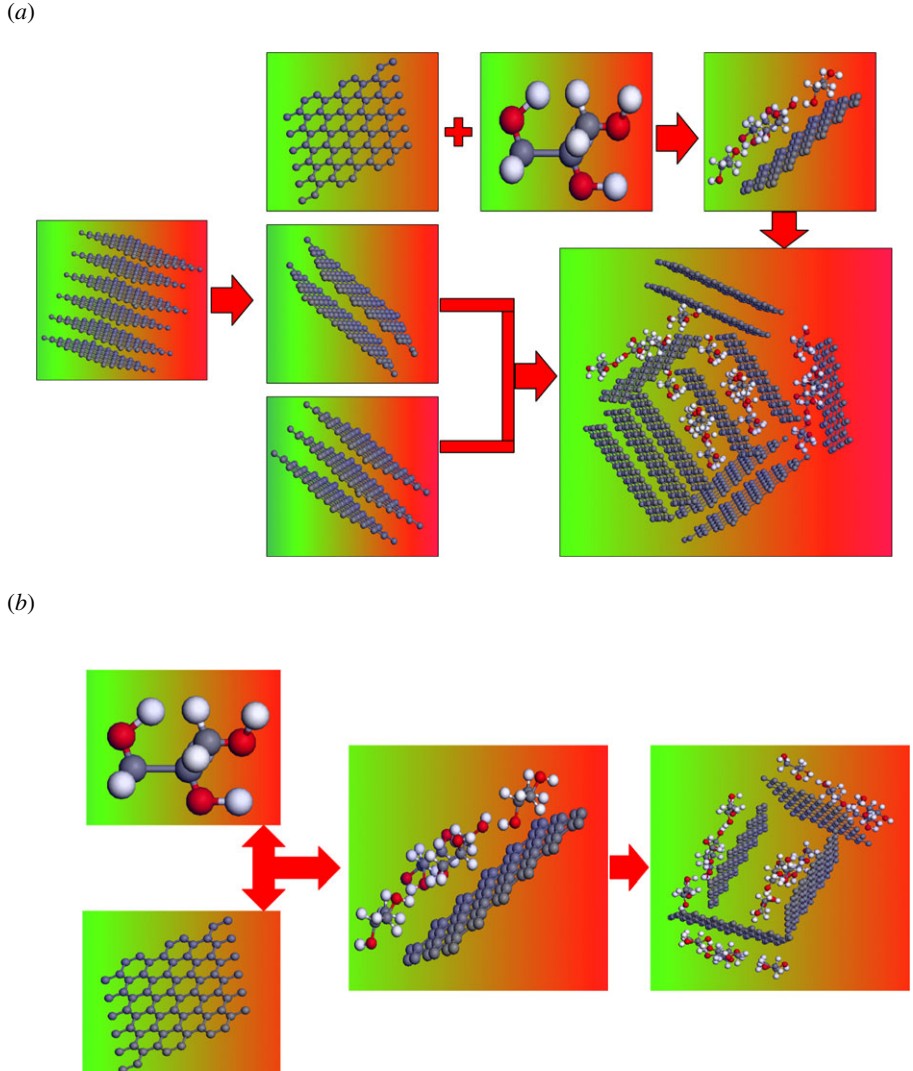

(*b*)

**Figure 8.** Adjust mechanism of intermolecular hydrogen bond. (*a*) The adjust mechanism of intermolecular hydrogen bond by graphite. The different layers of graphite made it play different roles, which increased its spatial confinement effect. (*b*) The adjust mechanism of intermolecular hydrogen bond by graphene. Its single-layer structure made its space-limiting action mainly depend on its own amount. Its ability to adjust the intermolecular hydrogen bond was weaker than that of graphite.

graphene interacts with glycerol molecules, which reduced the size of the newly formed glycerol cluster. These new molecular clusters were isolated more thoroughly by a larger volume of multi-layer graphite through space confinement, which greatly reduced the probability of intermolecular hydrogen bonding between clusters. The number of intermolecular hydrogen bonds was greatly reduced, which resulted to reduce the viscosity of glycerol. This also made the viscosity-reducing effect of graphite stronger than that of graphene, which was in accordance with the results in figure 3.

## 4. Conclusion

The intermolecular hydrogen bonds were adjusted by graphite and graphene, which resulted to decrease the viscosity of glycerol. Through experiment and computer simulation analyses, the hydrogen bond network structure of glycerol was destroyed, and the number of intermolecular hydrogen bonds was decreased. The different layered structure of graphite and graphene made their ability to adjust mechanism to intermolecular hydrogen bond and impact in the viscosity were different. The ability of graphite was affected by its additive amount and multi-layer structure, while that of graphene mainly depended on its additive amount. The effect of graphite (5.68%) is better than that of graphene (2.93%). In this paper, a new physical way was introduced to adjust the intermolecular hydrogen bond to reduce the viscosity of polyhydroxy liquid. At the same time, this work reveals the potential

interplay between nanomaterials and hydroxyl liquids, which will contribute to the field of solid–liquid coupling lubrication.

This paper found that graphite and graphene could be used as a solid nano-thinning agent with stable properties, and it was itself a good solid lubricant. But the side effect of them could destroy the integrity of the lubricating film on the surface of the friction pair and increase the friction coefficient. Its thinning effect was currently limited to liquids containing intermolecular hydrogen bonds. For the lubricating oil with a complex composition system, its function of thinning also remains to be proved.

Ethics. We declare that the work submitted for the publication is original, has not been published elsewhere, accepted for publication elsewhere or under editorial review for publication elsewhere; and that all the authors mutually agree with its content and have approved the paper for release and submission. The manuscript does not contain experiments using animals. At the same time, the manuscript does not contain human studies.

Data accessibility. The data file has been uploaded to the Dryad Digital Repository: https://doi.org/10.5061/dryad. 9cnp5hqjh [22].

Authors' contributions. Y.Y. carried out the molecular laboratory work and drafted the manuscript; G.Z. participated in the design of the study; X.X. collected field data; P.Z. carried out the statistical analyses; L.M. coordinated the study and helped draft the manuscript. All authors gave final approval for publication.

Competing interests. We declare we have no competing interests.

Funding. The work was financially supported by the National Natural Science Foundation of China (51675297, 51527901), China Postdoctoral Science Foundation (2019M650656) and Tianjin Natural Science Foundation (19JCQNJC04300).

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
