## [Peer Review File · Royal Society Open Science]

Review History

RSOS-210565.R0 (Original submission)

Review form: Reviewer 1

Is the manuscript scientifically sound in its present form?

No

Are the interpretations and conclusions justified by the results?

No

Is the language acceptable?

Yes

Do you have any ethical concerns with this paper?

No

Have you any concerns about statistical analyses in this paper?

No

Recommendation?

Major revision is needed (please make suggestions in comments)

Comments to the Author(s)

Reviewer Comment For Author:

This manuscript titled as “Intermolecular hydrogen bond ruptured by graphite with different lamellar number” deals with both experimental and theoretical aspects of view.

In my opinion the present manuscript should be revised by the following points which improve its quality and readability, make it suitable for publication in the Royal Society Open Science journal:

- The most important part is about enhancing clearness of the theoretical data which have been neglected by the authors, not reported in some table. It seems that authors have mostly relied on experimental data. It should be explored the results in more depth to clarify how computer simulation supported some of experimental data, to show to what extent it confirms the claim about intermolecular interaction in this work. Otherwise there is no clear evidence how the authors have handled it by computer modeling,

-The manuscript can be organized better, written in more acceptable grammatical form and avoid to use some similar sentences in manuscript specially in discussion. Also for two times it is referred to the goal of work in different parts of introduction which it is recommended to be mentioned once at the end of introduction.

Review form: Reviewer 2

Is the manuscript scientifically sound in its present form?

Yes

Are the interpretations and conclusions justified by the results?

Yes

Is the language acceptable?

No

Do you have any ethical concerns with this paper?

No

Have you any concerns about statistical analyses in this paper?

No

Recommendation?

Accept with minor revision (please list in comments)

Comments to the Author(s)

Yanchao et al. studied of the intermolecular HB ruptured by graphite with different lamellar number. Authors presented a method that the graphite and graphene are added into the glycerol, resulting in the intermolecular HBs were adjusted. The result showed that the spatial limiting action of graphite or graphene was the main reason for breaking the intermolecular HB structure. The work is interesting, and should be published on the Royal Society Open Science after a minor revision.

1. Please give a more detailed description for the different viscosity of the mixed fluid with different amounts of graphite and graphene.

2. The possible side effect of graphite and graphene on thinning of glycerol should be discussed.
3. The abstract need to be improved. Such as "For purpose of investigating this effect, we try to break intermolecular hydrogen bonds by some means. Based on this, we present a physical method to modulate intermolecular hydrogen bonds for not changing the physical-chemical properties of materials". Authors maybe want to present "we presented a method to modulate the intermolecular hydrogen bonds, but not changing the physical-chemical properties of materials." Some sentences are redundant.
4. The English language should be polished.

Decision letter (RSOS-210565.R0)

Dear Dr Yin:

Title: Intermolecular hydrogen bond ruptured by graphite with different lamellar number
Manuscript ID: RSOS-210565

Thank you for submitting the above manuscript to Royal Society Open Science. On behalf of the Editors and the Royal Society of Chemistry, I am pleased to inform you that your manuscript will be accepted for publication in Royal Society Open Science subject to minor revision in accordance with the referee suggestions. Please find the reviewers' comments at the end of this email.

The reviewers and handling editors have recommended publication, but also suggest some minor revisions to your manuscript. Therefore, I invite you to respond to the comments and revise your manuscript.

Because the schedule for publication is very tight, it is a condition of publication that you submit the revised version of your manuscript before 30-Jun-2021. Please note that the revision deadline will expire at 00.00am on this date. If you do not think you will be able to meet this date please let me know immediately.

- 1) A text file of the manuscript (tex, txt, rtf, docx or doc), references, tables (including captions) and figure captions. Do not upload a PDF as your "Main Document".

- 2) A separate electronic file of each figure (EPS or print-quality PDF preferred (either format should be produced directly from original creation package), or original software format)
- 3) Included a 100 word media summary of your paper when requested at submission. Please ensure you have entered correct contact details (email, institution and telephone) in your user account
- 4) Included the raw data to support the claims made in your paper. You can either include your data as electronic supplementary material or upload to a repository and include the relevant doi within your manuscript
- 5) All supplementary materials accompanying an accepted article will be treated as in their final form. Note that the Royal Society will neither edit nor typeset supplementary material and it will be hosted as provided. Please ensure that the supplementary material includes the paper details where possible (authors, article title, journal name).

Kind regards,
Dr Laura Smith
Publishing Editor, Journals

On behalf of the Subject Editor Professor Anthony Stace and the Associate Editor Dr Dattatray Late.

RSC Associate Editor:
Comments to the Author:
Accept with minor revisions

RSC Subject Editor:
Comments to the Author:
(There are no comments.)

Reviewer comments to Author:

Reviewer: 1

Comments to the Author(s)

Reviewer Comment For Author:

This manuscript titled as “Intermolecular hydrogen bond ruptured by graphite with different lamellar number” deals with both experimental and theoretical aspects of view.

In my opinion the present manuscript should be revised by the following points which improve its quality and readability, make it suitable for publication in the Royal Society Open Science journal:

- The most important part is about enhancing clearness of the theoretical data which have been neglected by the authors, not reported in some table. It seems that authors have mostly relied on experimental data. It should be explored the results in more depth to clarify how computer simulation supported some of experimental data, to show to what extent it confirms the claim about intermolecular interaction in this work. Otherwise there is no clear evidence how the authors have handled it by computer modeling,

-The manuscript can be organized better, written in more acceptable grammatical form and avoid to use some similar sentences in manuscript specially in discussion. Also for two times it is referred to the goal of work in different parts of introduction which it is recommended to be mentioned once at the end of introduction.

Reviewer: 2

Comments to the Author(s)

Yanchao et al. studied of the intermolecular HB ruptured by graphite with different lamellar number. Authors presented a method that the graphite and graphene are added into the glycerol, resulting in the intermolecular HBs were adjusted. The result showed that the spatial limiting action of graphite or graphene was the main reason for breaking the intermolecular HB structure. The work is interesting, and should be published on the Royal Society Open Science after a minor revision.

1. Please give a more detailed description for the different viscosity of the mixed fluid with different amounts of graphite and graphene.
2. The possible side effect of graphite and graphene on thinning of glycerol should be discussed.
3. The abstract need to be improved. Such as “For purpose of investigating this effect, we try to break intermolecular hydrogen bonds by some means. Based on this, we present a physical method to modulate intermolecular hydrogen bonds for not changing the physical-chemical properties of materials”. Authors maybe want to present “we presented a method to modulate the intermolecular hydrogen bonds, but not changing the physical-chemical properties of materials.” Some sentences are redundant.
4. The English language should be polished.

Author's Response to Decision Letter for (RSOS-210565.R0)

See Appendix A.

RSOS-210565.R1 (Revision)

Review form: Reviewer 2

Is the manuscript scientifically sound in its present form?

Yes

Are the interpretations and conclusions justified by the results?

Yes

Is the language acceptable?

Yes

Do you have any ethical concerns with this paper?

No

Have you any concerns about statistical analyses in this paper?

No

Recommendation?

Accept as is

Comments to the Author(s)

I think authors answer all my questions, and improve the paper. It can be published.

Decision letter (RSOS-210565.R1)

Dear Dr Yin:

Title: Intermolecular hydrogen bond ruptured by graphite with different lamellar number
Manuscript ID: RSOS-210565.R1

It is a pleasure to accept your manuscript in its current form for publication in Royal Society Open Science. The chemistry content of Royal Society Open Science is published in collaboration with the Royal Society of Chemistry.

On behalf of the Subject Editor Professor Anthony Stace and the Associate Editor Dr Dattatray Late.

RSC Associate Editor:
Comments to the Author:
Accept as is

RSC Subject Editor:
Comments to the Author:
(There are no comments.)

Reviewer(s)' Comments to Author:
Reviewer: 2

Comments to the Author(s)
I think authors answer all my questions, and improve the paper. It can be published.

Appendix A

Ph.D Yanchao Yin

Biological Engineering Technology Innovation Center of Shandong Province

Heze Branch, Qilu University of Technology (Shandong Academy of Sciences), Heze, 274000, Shandong, P.R. China.

Phone: +86 18810113957; E-mail: yycbm@163.com

Dear editor:

We would like to resubmit the revised manuscript entitled “*Intermolecular hydrogen bond ruptured by graphite with different lamellar number*” for consideration by *Royal Society Open Science*. We would like to thank the reviewers for thoroughly reviewing our manuscript and making many thoughtful comments. We have added significant new data, described in detail below, and revised the manuscript to address reviewers’ comments. Here are our point-by-point responses:

Reviewer #1:

Comment 1:

The most important part is about enhancing clearness of the theoretical data which have been neglected by the authors, not reported in some table. It seems that authors have mostly relied on experimental data. It should be explored the results in more depth to clarify how computer simulation supported some of experimental data, to show to what extent it confirms the claim about intermolecular interaction in this work. Otherwise there is no clear evidence how the authors have handled it by computer modeling.

Response:

In order to enhance the clarity of the theoretical data, table 1 was added to explain the change of the internal chemical bond length of glycerol molecules before and after adsorption on the surface of graphene.

Table 1 was added into page 10(red font).

“**Table 1. The length of C-O bond before and after adsorption.**”

Type of bond	Bond length before adsorption	Bond length after adsorption
C-O	1.421 Å	1.402 Å
C-O	1.452 Å	1.393 Å
C-O	1.422 Å	1.405 Å

Comment 2:

The manuscript can be organized better, written in more acceptable grammatical form

and avoid to use some similar sentences in manuscript specially in discussion. Also for two times it is referred to the goal of work in different parts of introduction which it is recommended to be mentioned once at the end of introduction.

Response:

According to the editor's opinion, lines 9-10 on page 9 (All this resulted that the thinning action of graphite was better than that of graphene in Figure 3.) and lines 12-14 on page 12 (This also made the viscosity-reducing effect of graphite stronger than that of graphene, which was in accordance with the results in Figure 3.) of the manuscript were repeatedly described, and the part repeatedly described on page 9 had been deleted.

Reviewer #2:

Comment 1:

Please give a more detailed description for the different viscosity of the mixed fluid with different amounts of graphite and graphene.

Response:

The more detailed description for the different viscosity of the mixed fluid with different amounts of graphite and graphene was added into page 6-7(red font).

“The rheology curve of the mixtures (25 °C) is shown in Figure 3a and 3b. As shown in Figure 3a, the viscosity of the mixture was lower than that of pure glycerol. Initial, when the amount of graphite was the largest ($x = 500$), the graphite was in a state of uniform dispersion in glycerol, and the ability to modulate the intermolecular hydrogen bond of glycerol was the strongest. With the decrease of adding amount, the modulation effect was weakened. Because of the large specific surface energy and the occurrence of small-scale agglomeration, when the amount of glycerol is reduced to $x=1000/1250/1500$, its ability to modulate the intermolecular hydrogen bond of glycerol was basically the same. It showed that the thinning action of graphite was more dependent on its mass. Its action had been greatly weakened at $x=1000$. Continuing to reduce its mass, the magnitude of viscosity reduction would not change significantly.

As shown in Figure 3b, the thinning action of graphene was also obvious. With the decrease of the amount of graphene, the change law of viscosity of glycerol was different from that of graphite. The surface of polar graphene was easier to adsorb hydrocarbons. The more hydrocarbons adsorbed, the lower the surface energy, which reduced the agglomeration phenomenon. Compared with graphite, the number of monolayer graphene was much higher than that of graphite at the same amount. When the amount of graphene was the largest ($y = 500$), its aggregation in glycerol was serious, and its adsorption capacity for glycerol was weak. With the decrease of the amount of graphene, when the amount of graphene was reduced to $y = 1000$, the distribution of graphene in glycerol was more dispersed, and the adsorption of glycerol on each graphene reached saturation state. At this time, the graphene adsorbed by glycerol had lower specific surface energy. The aggregation phenomenon was reduced, and the ability to modulate the intermolecular hydrogen bond of glycerol was the strongest. However, with the decrease of the addition amount($y=1250/1500$), the modulation ability was weakened. This introduced that the thinning action of graphene had an optimal value, which could greatly reduce the viscosity of glycerol.”

Comment 2:

The possible side effect of graphite and graphene on thinning of glycerol should be discussed.

Response:

The discussion of the possible side effect of graphite and graphene on thinning of glycerol was added into page 13(red font).

“This paper found that graphite and graphene could be used as a solid nano-thinning agent with stable properties, and it was itself a good solid lubricant. But the side effect of them could destroy the integrity of the lubricating film on the surface of the friction pair and increase the friction coefficient. Its thinning effect was currently limited to liquids containing intermolecular hydrogen bonds. For the lubricating oil with complex composition system, its function of thinning also remains to be proved.”

Comment 3:

The abstract need to be improved. Such as “For purpose of investigating this effect, we try to break intermolecular hydrogen bonds by some means. Based on this, we present a physical method to modulate intermolecular hydrogen bonds for not changing the physical-chemical properties of materials”. Authors maybe want to present “we presented a method to modulate the intermolecular hydrogen bonds, but not changing the physical-chemical properties of materials.” Some sentences are redundant.

Response:

According to the editor's opinion, the redundant sentences (For purpose of investigating this effect, we try to break intermolecular hydrogen bonds by some means.) have been deleted from the abstract.

Comment 4:

The English language should be polished.

Response:

Ok, I will improve my English writing level in my future work.

Thank you for your consideration of our manuscript.

Ph.D Yanchao Yin

Biological Engineering Technology Innovation Center of Shandong Province

Heze Branch, Qilu University of Technology (Shandong Academy of Sciences), Heze, 274000, Shandong, P.R. China.

Phone: +86 18810113957; E-mail: yycbm@163.com